# Alginate microencapsulation of an attenuated O-antigen mutant of *Francisella tularensis* LVS as a model for a vaccine delivery vehicle

**Kelly C. Freudenberger Catanzaro**[1¤], **Kevin K. Lahmers**[1], **Irving C. Allen**[1], **Thomas J. Inzana**[1,2]*

**1** Department of Biomedical Sciences and Pathobiology, Virginia-Maryland College of Veterinary Medicine, Virginia Tech, Blacksburg, Virginia, United States of America, **2** College of Veterinary Medicine, Long Island University, Brookville, New York, United States of America

¤ Current address: BluePearl Speciality & Emergency Pet Hospital, Cary, North Carolina, United States of America
* Thomas.Inzana@liu.edu

## Abstract

*Francisella tularensis* is the etiologic agent of tularemia and a Tier I Select Agent. Subspecies *tularensis* (Type A) is the most virulent of the four subspecies and inhalation of as few as 10 cells can cause severe disease in humans. Due to its niche as a facultative intracellular pathogen, a successful tularemia vaccine must induce a robust cellular immune response, which is best achieved by a live, attenuated strain. *F. tularensis* strains lacking lipopolysaccharide (LPS) O-antigen are highly attenuated, but do not persist in the host long enough to induce protective immunity. Increasing the persistence of an O-antigen mutant may help stimulate protective immunity. Alginate encapsulation is frequently used with probiotics to increase persistence of bacteria within the gastrointestinal system, and was used to encapsulate the highly attenuated LVS O-antigen mutant WbtI$_{G191V}$. Encapsulation with alginate followed by a poly-L-lysine/alginate coating increased survival of WbtI$_{G191V}$ in complement-active serum. In addition, BALB/c mice immunized intraperitoneally with encapsulated WbtI$_{G191V}$ combined with purified LPS survived longer than mock-immunized mice following intranasal challenge. Alginate encapsulation of the bacteria also increased antibody titers compared to non-encapsulated bacteria. These data suggest that alginate encapsulation provides a slow-release vehicle for bacterial deposits, as evidenced by the increased antibody titer and increased persistence in serum compared to freely suspended cells. Survival of mice against high-dose intranasal challenge with the LVS wildtype was similar between mice immunized within alginate capsules or with LVS, possibly due to the low number of animals used, but bacterial loads in the liver and spleen were the lowest in mice immunized with WbtI$_{G191V}$ and LPS in beads. However, an analysis of the immune response of surviving mice indicated that those vaccinated with the alginate vehicle upregulated cell-mediated immune pathways to a lesser extent than LVS-vaccinated mice. In

**Data Availability Statement:** All relevant data are within the paper and its Supporting information files.

**Funding:** K F-C was supported by the Stamps Family Charitable Foundation (no fund number). TJI was supported by the Tyler J. and Frances F. Young Foundation, and the Virginia-Maryland College of Veterinary Medicine (no fund number).

**Competing interests:** The authors have declared that no competing interests exist.

summary, this vehicle, as formulated, may be more effective for pathogens that require predominately antibody-mediated immunity.

## Introduction

*Francisella tularensis* is a gram-negative, coccobacillus that causes the zoonotic disease tularemia. *F. tularensis* species are considered Tier I Select Agents due to the potential use of this bacterium as a bioweapon [1]. This classification is due to the degree of virulence, low infectious dose, and ease of aerosol dispersal of the bacterium [1, 2]. No vaccine to prevent tularemia is currently licensed in the United States, including the Live Vaccine Strain (LVS) [3]. The LVS strain was developed from serial passages of a wildtype *F. tularensis* subspecies *holarctica* (Type B) strain in the mid-20th century and is attenuated compared to wild-type strains [4]. However, LVS is still virulent in mice and immunocompromised populations, and has questionable protective efficacy against respiratory challenge [5–7]. These concerns have regulated LVS to the status of a laboratory model strain for *F. tularensis* and the standard of comparison for potential tularemia vaccines. Any possible new tularemia vaccine candidate should be as efficacious or more so than LVS, especially against respiratory challenges, have a stable genetic composition, and must be safer in immunocompromised individuals [3].

Deletion of the lipopolysaccharide (LPS) O-antigen significantly attenuates even highly virulent type A *F. tularensis* strains by several logs [8–14]. A genetically stable strain lacking the O-antigen would be unlikely to revert to a virulent state *in vivo* and would be a safe vaccine candidate for immunocompromised individuals. Although O-antigen mutants are highly serum sensitive, in contrast to the wildtype they do not persist long in the host, and are inadequate at inducing protective immunity against virulent strains [10, 12, 14, 15]. Li *et al*. [14] created the O-antigen mutant WbtI$_{G191V}$, a point mutant and grey variant of LVS. The point mutation is in the gene *wbtI* that encodes a glycosyl transamine/perosamine synthetase, required for elongation of the polysaccharide O-antigen portion of the LPS. WbtI$_{G191V}$ is sensitive to killing in 10% fresh serum in less than 60 min., and unable to persist in the host to establish an active infection [14]. When used to immunize mice WbtI$_{G191V}$ is protective against a low-dose intraperitoneal challenge with virulent LVS, but ineffective against a high dose challenge [14]. Twine *et al*. [16] reported similar results of partial and route-dependent protection with LVS mutants lacking WbtC or KdtA. Although attenuating LVS, the loss of the O-antigen may hinder the host from developing a fully protective immune response against *F. tularensis* infection due to the lack of persistence of the bacteria *in vivo*. In addition, antibodies to the LPS O-antigen, which may supplement protective immunity [11], are absent. Strategies to improve the efficacy of partially protective vaccine candidates have been employed. A common strategy is repeated immunizations. Protection of WbtI$_{G191V}$ against a wild-type challenge is greater when a booster immunization is used [14], compared to a single dose immunization [16]. However, the booster immunization still only provides incomplete protection against highly virulent type A strains, indicating that a more novel approach is necessary.

Liposomes have long been used as a delivery vehicle for proteins and other small antigens [17], but are too small to encapsulate whole bacteria. Alginate is a carbohydrate polymer that crosslinks when introduced to a solution of calcium forming a stable, relatively large bead. When bacteria are mixed with alginate the bacteria are trapped within the polymerized matrix following the introduction of calcium. This technique has been used to protect probiotics for

passage through acidic environments of the gastrointestinal track [18–21]. *Lactobacillus* spp. encapsulated in alginate can survive in simulated gastric juices for extended periods of time compared to non-encapsulated *Lactobacillus* cells [22–24]. In theory, the alginate barrier isolates cells within the microcapsule from the surrounding environment, increasing persistence. This same approach could be applied to the protection of highly attenuated bacteria from the host innate immune system. Microencapsulation of live bacteria with alginate has been used to increase host immunity against highly attenuated *Brucella* strains [25–27]. Attenuated *Brucella melitensis* encapsulated in alginate microspheres induces cell-mediated and antibody-mediated immune responses that are greater than freely suspended cells, and was effective at reducing the bacterial burden in mouse tissues [25]. Alginate encapsulation has also been effectively used to stimulate protection against *Yersinia ruckeri* in rainbow trout [28], *Piscirickettsia salmonis* in salmon [29], and *Bordetella pertussis* in mice [30]. Therefore, in theory, when used as a vaccine vehicle alginate encapsulation should prevent the immediate clearance of the attenuated strain, allowing the agent to be slowly released over an extended period of time, and continuously stimulate the immune system.

In applying this principle to the attenuated strain $WbtI_{G191V}$, we predicted that alginate microencapsulation would increase the persistence of $WbtI_{G191V}$ *in vivo*. This increased persistence should act as a continuous booster, and help stimulate an appropriate and protective immune response to protect the host against a virulent, intranasal challenge.

## Methods and materials

### Ethics statement

All proposals involving research utilizing living vertebrates are reviewed by the Virginia Tech Institutional Animal Care and Use Committee (IACUC) in accordance with the federal Animal Welfare Act, the "Public Health Service Policy on Humane Care and Use of Laboratory Animals," and the "Guide for Care and Use of Laboratory Animals." Virginia Tech's Animal Welfare Assurance number is A-3208-01 with an expiration of 7-31-2021. All animals used in this study were approved by the Virginia Tech IACUC under the approved protocol 17–242.

### Bacterial strains and growth conditions

The bacterial strains used in this study included the attenuated O-antigen mutant *F. tularensis* LVS $WbtI_{G191V}$ [14], the mouse-virulent wildtype strain *F. tularensis* LVS, and *F. tularensis* LVS expressing green fluorescent protein (GFP) [31]. *Francisella* strains were grown on chocolate brain heart infusion agar (BD) containing 0.1% cysteine (CBHI-C) at 37°C with 5% $CO_2$. Broth cultures of *Francisella* strains were grown in brain heart infusion broth containing 0.1% cysteine (BHI-C) with shaking (175–200 rpm) at 37°C. The GFP-expressing LVS strain was grown with the addition of 10 μg/ml of kanamycin in the growth medium.

### Encapsulation of *Francisella* strains in alginate beads

A 48-hour culture of *F. tularensis* was scraped off CBHI-C agar plates and suspended in morpholinepropanesulfonic acid buffer (MOPS: 10mM MOPS, 0.85% NaCl, pH 7.4). The bacteria were pelleted by centrifugation at 10,000 x *g*, the supernatant was removed, and fresh MOPS was added to resuspend the pellet. This washing step was repeated once. The final volume was adjusted to yield a suspension of $1 \times 10^9$ colony forming units (CFU)/ml, determined spectrophotometrically, which was confirmed by viable plate count. This suspension was then diluted into 1.8% isotonic sodium alginate (Buchi, Switzerland) to obtain a final concentration between $1 \times 10^6$ and $1 \times 10^8$ CFU/ml in 1.2% alginate for nozzle sizes of 80–120 μm or 1.5%

**Table 1. Parameters for production of alginate beads containing $1\times10^8$ or fewer CFU/ml of *F. tularensis* bacteria.**

| Nozzle Size (um) | Final Alginate Concentration (%) | Flow Rate (ml/min) | Frequency (Hertz) | Electrode (Volts) |
|---|---|---|---|---|
| 80 | 1.2 | 1.20 | 2700 | 450 |
| 120 | 1.2 | 1.40 | 2500 | 500 |
| 150 | 1.5 | 2.50 | 1500 | 750 |
| 200 | 1.5 | 4.00 | 1200 | 1100 |

alginate for nozzle sizes 150–200 μm. Purified *F. tularensis* LVS LPS was included at a concentration of 100 μg/ml in the alginate-bacterial suspension of specific bead preparations. Alginate beads that contained *F. tularensis* with or without LPS were produced as previously described with modification [25]. Briefly, the alginate-bacterial suspension was loaded into a syringe and connected to the Buchi Encapsulator B-395 Pro (Buchi, Switzerland). The encapsulation parameters for the various nozzle sizes are described in Table 1. The encapsulator extruded the suspension through the attached nozzle into sterile calcium chloride buffer (CaCl$_2$: 100 mM CaCl$_2$, 10 mM MOPS pH 7.4), and the mixture was slowly stirred for 15 to 30 minutes. The CaCl$_2$ buffer was removed, leaving only a small amount of liquid to prevent the beads from sticking. The beads were then washed twice with MOPS buffer for 5 minutes each. If necessary, a portion of the beads were then removed, diluted, and cultured to determine bacterial numbers, and for use in survival and animal studies. Beads removed at this point were designated as alginate beads. Three independent experiments per condition (nozzle size, alginate concentration, flow rate, frequency, and electrode voltage) were carried out (Table 1).

The beads were then slowly stirred in a poly-L-lysine solution (PLL: 0.05% PLL molecular weight 30–70 kDa (Sigma P2636) in CaCl$_2$ buffer, filter sterilized) for 15 minutes at room temperature (approximately 20–22 ˚C) to create the coating layer around the alginate bead. Following incubation, the beads were washed twice in MOPS buffer for 5 minutes each. A 0.03% solution of sodium alginate was added and the beads were slowly stirred for 5 minutes to create the final shell. Completed beads were washed in MOPS buffer twice before use. These beads coated twice with alginate and PPL were designated Alginate-PLL-Alginate (APA) beads. Bead formulations are presented in Table 2. Three independent experiments per condition were carried out.

## Assessment of bacterial concentration within alginate beads

Initially, the number of WbtI$_{G191V}$ bacterial cells that could be accommodated within the alginate beads was determined. Approximately 250 μl of alginate beads in 250 μl of MOPS buffer were added to 9.5 ml of solubilization solution (50 mM sodium citrate, 0.45% NaCl, 10 mM MOPS, pH 7.4). The suspension was incubated at 37˚C with shaking (150 rpm) for 10 minutes until the beads were dissolved. The suspension was serially diluted and the concentration of bacteria within the beads was determined by viable plate count. A portion of the original alginate-bacterial suspension was used as a control to determine the starting bacterial

**Table 2. Characteristics of alginate beads produced throughout the study.**

| Bead Designation | Bead Core | Bead Coating | Purpose |
|---|---|---|---|
| Alginate LVS GFP Beads | *F. tularensis* LVS GFP strain, $1\times10^8$ CFU/ml | None | Microscopy |
| Alginate WbtI Beads | *F. tularensis* LVS WbtI strain | None | Microscopy, Bacterial viability, *in vivo* |
| APA WbtI Beads | *F. tularensis* LVS WbtI strain | PLL and Alginate | Microscopy, Bacterial viability, *in vivo* |
| APA WbtI and LPS Beads | *F. tularensis* LVS WbtI strain and 100 ug/ml LVS LPS | PLL and Alginate | *In vivo* |

concentration before encapsulation in $CaCl_2$. The encapsulation efficiency of a particular sample was determined to be the bacterial concentration present in 1 ml of beads, divided by the initial bacterial concentration in 1 ml of alginate, multiplied by 100. Three independent experiments per condition were carried out.

## Characterization of alginate beads

Alginate beads were assessed via light microscopy to determine their sphere morphology and shape for different preparations. For each preparation, variation in the size of 30 beads were determined (ten beads were measured from 3 independent experiments per condition.) The diameter of each bead was measured on an Olympus BX41 microscope using the Olympus DP Controller software (Olympus, Waltham, Massachusetts) and the mean values were determined.

## Bacterial survival in complement active serum

Alginate beads and APA beads containing $WbtI_{G191V}$ were incubated in 10% fresh guinea pig serum for 0 hours, 1 hour, 24 hours, and 48 hours. Freely suspended $WbtI_{G191V}$ and freely suspended LVS cells were used to evaluate percent killing by the serum as positive and negative controls, respectively. Aliquots of the alginate beads, freely suspended $WbtI_{G191V}$, and freely suspended LVS were harvested at each time point and subjected to dissolution and viable plate count, as described above. CFUs were determined for each time point and compared to the CFU count from 0 hours. Three independent experiments were run in duplicate per experiment.

APA beads could not be dissolved in sodium citrate due to the covalent crosslinking of PLL and alginate. Instead, after incubation in complement-active serum these beads were washed and suspended in BHI-C broth to determine if bacteria could grow or did not grow from the beads. Alginate beads, freely suspended $WbtI_{G191V}$, and freely suspended LVS also underwent this same procedure, including controls.

## Immunization and challenge of mice

An equal number of female and male BALB/c mice 6–8 weeks old (Charles River Laboratories, Wilmington, MA) were housed in an AALAC-accredited ABSL-2 facility. Groups of 4 mice (2 male and 2 female) were used to assess the protective efficacy of APA beads containing $WbtI_{G191V}$ with or without purified LVS LPS against intranasal (IN) challenge. Mice were inoculated with either (i) phosphate buffered saline (controls; 2 mice), pH 7.2 (PBS), (ii) $10^7$ CFU of $WbtI_{G191V}$, (iii) $10^7$ CFU of $WbtI_{G191V}$ with 10 µg of purified LPS, (iv) $10^7$ CFU of $WbtI_{G191V}$ in APA beads, or (v) $10^7$ CFU of $WbtI_{G191V}$ combined with 10 µg of purified LPS in APA beads intraperitoneally (IP) or subcutaneously (SQ). A group of 4 mice were inoculated subcutaneously with $10^5$ CFU of virulent LVS as a comparative control for vaccine efficacy. A 25-guage needle was used for the IP and SQ immunizations. Mice were weighed and monitored for the presence of clinical signs twice daily for 2 weeks after immunizations. Mice were cheek bled 4 weeks after the immunizations. Collected blood was incubated at room temperature for approximately 15 minutes and was centrifuged at 10,000 x *g* to remove the clot. Serum was collected and stored to determine *Francisella*-specific antibody titers. Six weeks after immunizations, mice were anesthetized with 3–4% isoflurane and inoculated intranasally (IN) with $10^6$ CFU of LVS, which is a very high level challenge dose. All mice were monitored for 2 weeks and any mouse that became moribund was euthanized immediately with excess $CO_2$. Mice that survived for 2 weeks were euthanized by the same procedure. Liver, lung, and spleen tissues were harvested from all euthanized mice. Tissue samples were homogenized in PBS containing 1 mM $CaCl_2$ and 2 mM $MgCl_2$. The homogenized tissue suspensions were diluted

serially ten-fold and the bacterial concentration in weighed tissue was determined by viable plate count.

## Determination of anti-LVS titers

The relative level of *Francisella* LVS-specific antibodies in collected sera was measured by enzyme-linked immunosorbent assay (ELISA). An overnight plate culture of LVS was resuspended in PBS at a concentration of $10^9$ CFU/ml and heat-inactivated at 65˚C for 1 hour. Heat-inactivated LVS was diluted in carbonate buffer (50 mM, pH 9.6) to a concentration of $10^8$ CFU/ml. Ninety-six well Immulon plates (Thermo Scientific) were coated with 100 μl of heat-inactivated LVS in carbonate buffer and incubated at 4˚C overnight. The plates were washed 3 times with PBS containing 0.05% Tween 20 (PBST) and then blocked with 1% non-fat dry milk (NFDM) at 37˚C for 1 hour. The plates were washed once with PBST and serum samples were added in duplicate diluted 1:100, followed by two-fold serial dilutions. Plates were incubated at 37˚C for 1 hour and then washed 3 times with PBST. Aliquots of 100 μl of horseradish peroxidase-conjugated anti-mouse IgG (heavy and light chain; Jackson ImmunoResearch Laboratories, West Grove, PA) diluted in NFDM at a concentration of 1:1000 were added to the plate and incubated for 1 hour at 37˚C. The plate was washed an additional 3 times with PBST and the color developed with the TMB Substrate Kit (Thermo Fisher) per the manufacturer's instructions. The color reaction was terminated with 2M $H_2SO_4$ and the optical density was measured at 450 nm. Nonspecific reactivity was determined from the OD of control wells that included all reagents except bacterial cells. Antibody titers were defined as the reciprocal of the highest dilution of immune serum with an OD value that was 0.1 greater than the mean OD of similarly diluted serum from non-immunized mice. Serum from naïve mice and blank wells were used for comparisons.

## Innate and adaptive immune response profile of lung tissues

Fresh, frozen (-80 ˚C) lung tissue from mice that survived challenge were homogenized as previously described. Total RNA was isolated from each homogenized sample using the RNeasy Mini Kit (Qiagen) as described by the manufacturer and then pooled proportionately by immunization group. The RT$^2$ First Strand Kit was used to create cDNA per the manufacturer's instructions (Qiagen). Synthesized cDNA was used as the template for the RT$^2$ Profile PCR Array for Mouse Adaptive and Innate Immune Response (Qiagen) and run per the manufacturer's instructions on the Applied Biosystems 7500 Real-Time PCR System. Lung tissue from mice that had not been immunized or challenged were used to determine basal levels of each gene product. Ingenuity Pathway Analysis (IPA, Qiagen) was used to analyze the up- or down-regulation of the cellular immune pathway from each group.

## Histopathology of lung, liver, and spleen samples

Sections of lung, liver, and spleen were fixed in 10% neutral buffered formalin at the time of necropsy for each mouse (4 mice per IP and SQ groups and 3 to 4 samples/mouse/tissue, resulting in 42 samples analyzed, and read blindly). The samples went sent to the histopathology laboratory at the Virginia-Maryland College of Veterinary Medicine (VMCVM) for processing and staining. Briefly, fixed samples were embedded in paraffin, sectioned, and stained with hematoxylin and eosin (H&E) stain. Stained tissue sections were graded according to published criteria [32, 33]. Briefly, scores ranging from 0–3 were assigned to criteria that included the type of inflammation, location, and severity of cellular infiltrates. The tissues were evaluated 3 times by a single, American College of Veterinary Pathologists board-certified pathologist with the consensus score for each sample and category provided.

## Statistical analyses

Two-way ANOVAs followed by Tukey's *post-hoc* test were used to evaluate differences in characteristics (*i.e.* bead size, bacterial concentration, and efficiency) of produced alginate beads based on different production parameters, such as nozzle size and bacterial starting concentration. The Mantel-Cox log-rank test was used to compare the survival curves of the control and immunized mice following challenge. Kruskal-Wallis one-way ANOVA was used to evaluate the presence of significant differences in antibody titers post-immunization, bacterial loads after challenge, and histopathology scores of the various mouse groups. Dunn's multiple comparisons test or Tukey's post-hoc was used after completion of the previous test to identify specific statistical differences between immunized mouse groups and the control sham-immunized mice. Statistical analyses were determined using GraphPad Prism 8 (GraphPad Software Inc., La Jolla, CA).

## Results

### Sodium alginate bead characteristics

Prior to determining the optimal parameters for immunization, an LVS strain expressing GFP (LVS-GFP) was used to visualize the presence of viable bacteria within the beads after production (Fig 1). LVS-GFP was successfully encapsulated in alginate using an 80-μm nozzle on a Buchi Encapsulator B-395 Pro. The spheres were all less than 200 μm in diameter and relatively spherical, although occasional spheres exhibited a more tear-drop shape (not shown). Therefore, *F. tularensis* could effectively be encapsulated in alginate.

Variations in bead structure, size, and bacterial content were examined to determine the eventual optimal immune response to *F. tularensis*. Variations in the alginate beads were created to assess how additional structures to the beads, such as a coating or content, would

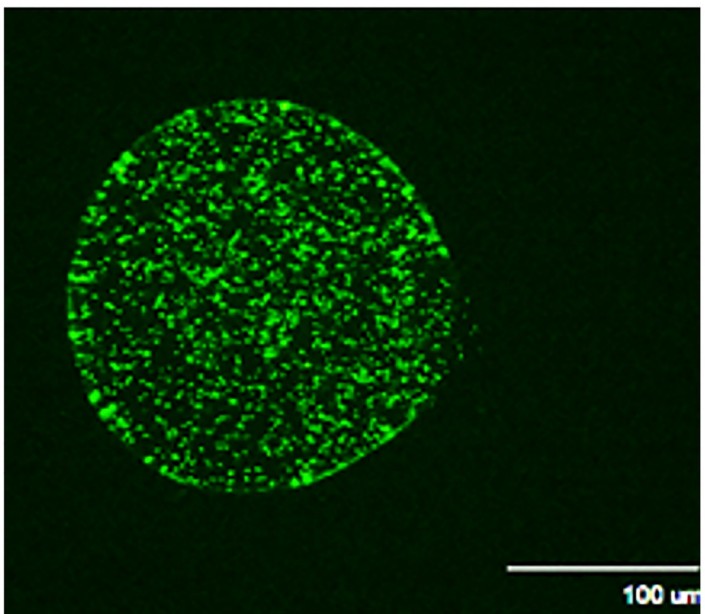

**Fig 1. Alginate bead containing GFP-expressing LVS.** Alginate beads were created that contained LVS expressing the fluorescent protein GFP. Only viable bacterial cells inside the generally spherical shaped capsule are visible by fluorescence microscopy.

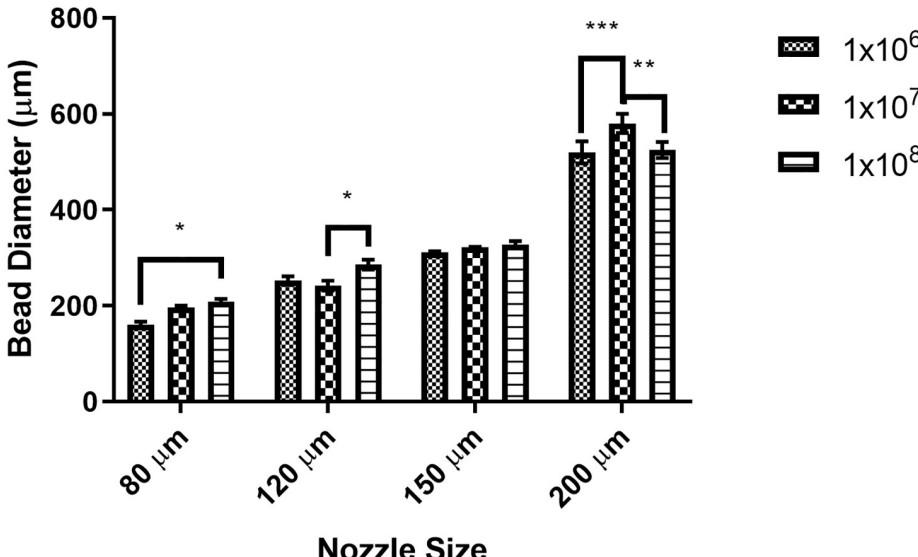

**Fig 2. Effect of production parameters on bead size.** Bead diameter was assessed for 30 beads produced using nozzle sizes from 80 μm to 200 μm and starting bacterial concentrations of 1x10⁶ to 1x10⁸ CFU/ml. Bead sizes statistically differed due to nozzle sizes ($p < 0.0001$) and due to starting bead concentration ($p = 0.0032$). The interaction between nozzle size and starting bacterial concentration was statistically important ($p = 0.0019$). Significant differences between starting cell concentration within nozzle groups were designated as: *: $p < 0.05$, **: $p < 0.01$, ***: $p < 0.001$.

contribute to bead formation and survival of the encapsulated bacteria. The beads were created with and without a PLL and alginate coating (Table 2). Alginate beads that were coated with PLL followed by a final layer of alginate were designated APA beads. As mentioned, while some beads exhibited a tear-drop form, those beads that received the PLL plus final alginate coating no longer exhibited the tear drop shape.

Beads were also produced with varying levels of bacterial cell concentrations. Previous protocols for live cell encapsulation using the Buchi Encapsulator B-395 Pro limited the starting cell concentration to approximately 1x10⁶ cells/ml [25]. Per the manual, a final concentration of less than 1x10¹⁰ non-human cells/ml was recommended [34]. Log increases in starting cell concentration were tested to determine if a final concentration of 1x10¹⁰ LVS cells/ml could be obtained and how an increase in cell concentration affected bead characteristics. A starting concentration of 1x10⁹ cells/ml resulted in the formation of rafts of alginate on the surface of the polymerization solution (not shown). This represented the cell concentration at which beads no longer formed, possibly due to an inadequate ratio of alginate to cells. At this point, all further assays were conducted with less than 1x10⁹ cells/ml in the starting solution.

The diameter of over 30 beads was measured from each microencapsulation run to assess the contribution of starting bacterial cell concentration and nozzle size to the bead diameter. The mean diameter of the beads was significantly affected by the nozzle size ($p < 0.0001$), starting bacterial concentration ($p = 0.0032$), and the interaction between the two ($p = 0.0019$) (Fig 2). The average size of beads across all starting bacterial concentrations produced with the 80-μm nozzle was approximately 190 μm in diameter, with the use of a starting concentration 1x10⁶ cells/ml. This concentration of bacteria resulted in a statistically significant size difference compared to using the highest concentration of 1x10⁸ cells/ml ($p = 0.0148$). Use of the 200-μm nozzle increased the bead diameter to approximately 550 μm, more than double the size of the beads produced with the 80-μm nozzle. Although there were statistically significant

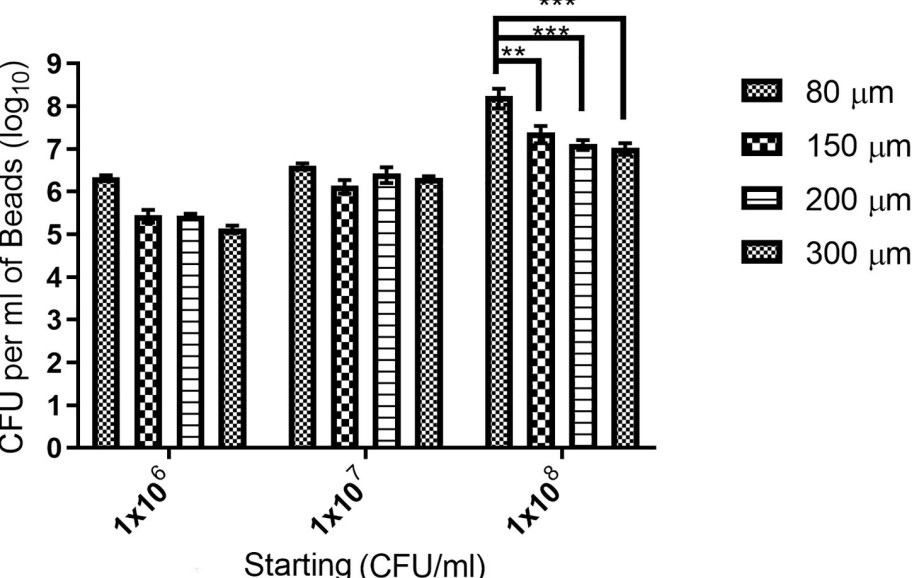

**Fig 3. Bacterial concentration within beads determined by the bacterial starting concentration.** The viable bacterial concentration in a milliliter of beads was determined by dissolution of beads in citrate solution and viable plate count. Bacterial concentration within beads was statistically significant depending on the starting bacterial concentration ($p = 0.0097$) and was statistically affected by the interaction of starting concentration and nozzle size ($p = 0.0206$). Nozzle size alone did not statistically affect the bacterial concentration of the beads ($p = 0.0646$). Differences between nozzle sizes within groups is signified as: **: $p < 0.01$, ***: $p < 0.001$.

differences in bead size when the initial bacterial cell concentrations varied, the main determinant of bead size was the nozzle size.

Dissolution of the alginate beads in buffered sodium citrate was performed to determine the concentration of viable bacteria/ml of alginate beads (Fig 3). This assay was performed only on alginate beads and not APA beads due to the covalent crosslinking between alginate and PLL that prevents dissolution. The concentration of viable bacteria in the APA beads was presumed to be the same as in their non-coated counterparts. The starting bacterial concentration significantly affected the final bacterial concentration ($p = 0.0097$), and the interaction between nozzle size and starting bacterial concentration significantly affected the final bacterial concentration per ml of alginate beads ($p = 0.0206$). No statistical differences were found between the final bacterial concentration and nozzle size when the starting concentration was $1\times10^6$ or $1\times10^7$ CFU/ml ($p > 0.9998$). When bacterial starting concentrations were increased to $1\times10^8$ CFU/ml, the final bacterial CFU/ml of beads decreased with the increased nozzle size (Fig 3). When a starting concentration $1\times10^8$ CFU/ml was used in conjunction with the 80-μm nozzle, the highest final concentration of bacteria within the beads was achieved.

The encapsulation efficiency of each batch of beads produced was classified as the proportion of bacterial cells released from alginate beads compared to the bacterial cells present in the starting alginate solution as a percent. Encapsulation efficiency was compared between all starting bacterial concentrations, the nozzle size, and the interaction of starting bacterial concentration with the nozzle size (Fig 4). Encapsulation efficiency greater than 100% (80 μm nozzle) was classified as approximately 100% efficiency, and therefore no standard error bars are shown. Nozzle size ($p = 0.0005$) and starting bacterial concentration ($p < 0.0001$) were the specific parameters that statistically affected the encapsulation efficiency. However, the interaction between nozzle size and bacterial concentration was not statistically significant ($p = 0.0943$). In general, as the

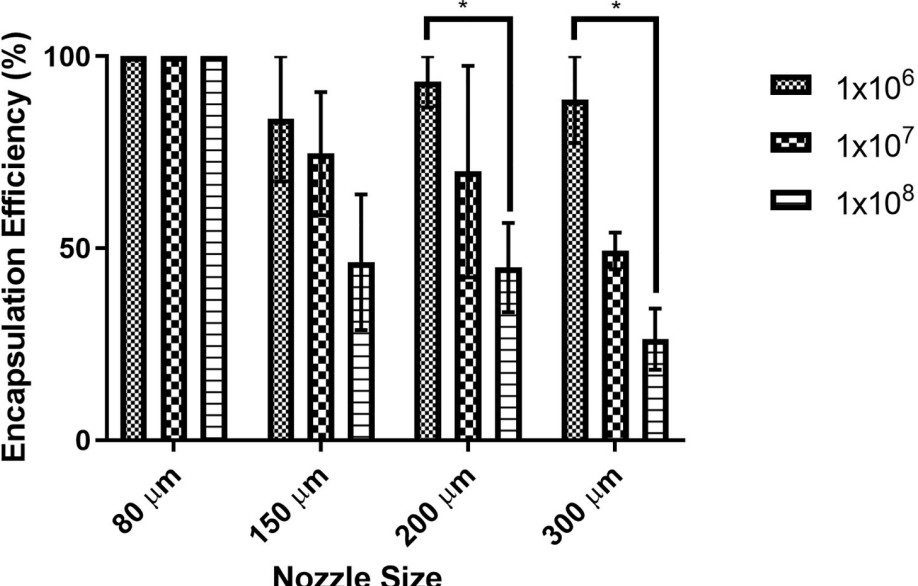

**Fig 4. Encapsulation efficiency significantly decreases with increasing starting bacterial concentration.**
Encapsulation efficiency was defined as the percentage of bacterial cells that became incorporated into the final alginate bead. Encapsulation efficiency decreased with increasing starting bacterial concentrations. Significant differences between starting bacterial concentrations within nozzle sizes are shown as: $^{*}$: $p < 0.05$.

starting bacterial concentration increased the encapsulation efficiency would decrease, with a smaller proportion of cells becoming entrapped in the alginate beads. This mirrors the decrease in the final bacterial concentration within the beads when using $1x10^8$ CFU/ml as the starting concentration (Fig 3). The encapsulation efficiency remained at approximately 100% for the smallest nozzle and was most variable for the largest nozzle size tested. Therefore, for all immunization studies, beads were created using the 80-μm nozzle and a starting concentration of $1x10^8$ CFU/ml. This combination produced one of the most efficient encapsulation rates and the highest concentration of bacterial cells within the final bead preparation.

Previous studies have shown that the LVS O-antigen deficient strain WbtI$_{G191V}$ is highly attenuated, and that up to $2.8 \times 10^7$ CFU given IP does not result in murine mortality [14]. Mice in this study only received 100 μl of beads containing WbtI$_{G191V}$ IP. Inoculations of mice with beads made using the 80-μm nozzle and a starting concentration of $1x10^8$ CFU/ml would contain an inoculation dose of less than $2.8x10^7$ CFU and therefore safe. Some mice that received $2.8x10^7$ CFU of WbtI$_{G197V}$ IP in the aforementioned study did exhibit clinical signs that resolved by 5 days post infection (PI).

### Encapsulation protects the serum-sensitive, O-antigen deficient mutant WbtI$_{G191V}$ from complement-mediated lysis

The O-antigen mutant WbtI$_{G191V}$ is serum-sensitive and is killed in approximately 1% of pre-colostral calf serum [14]. Encapsulation in alginate should create a barrier protecting the cells from complement proteins in the environment and increase survival time of WbtI$_{G191V}$. To test this hypothesis, WbtI$_{G191V}$ was encapsulated in alginate beads and in APA beads, and then incubated in complement-active guinea pig serum. After dissolving the alginate beads in sodium citrate buffer, survival of WbtI$_{G191V}$ in these bead formulations was compared to freely

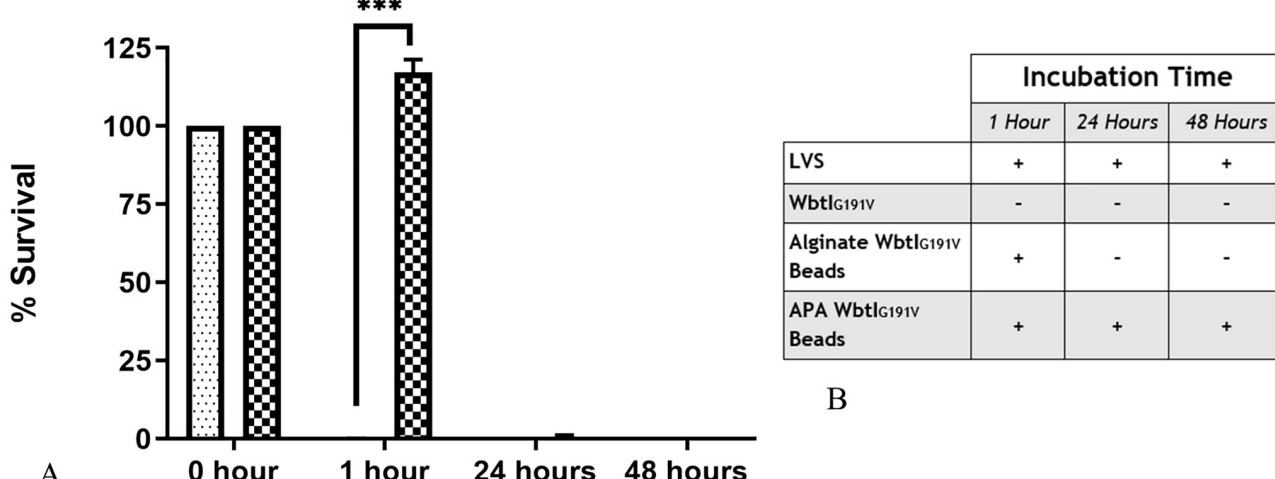

**Fig 5. Encapsulation increases survival of serum-sensitive WbtI$_{G191V}$ incubated in complement-active serum.** WbtI$_{G191V}$ was encapsulated in alginate or APA beads and then incubated in complement-active serum for 1 hour, 24 hours, and 48 hours. A: Viable plate counts were used to determine the percent survival of WbtI$_{G191V}$ freely suspended in serum (dotted bar) compared to WbtI$_{G191V}$ encapsulated in alginate beads (diagonal line bar). WbtI$_{G191V}$ encapsulated in alginate only survived significantly more than freely suspended WbtI$_{G191V}$ after 1 hour ($p < 0.001$, ***), and increased in number due to growth during that time. Viable WbtI$_{G191V}$ cells in alginate beads were detected after 24 hours of incubation in complement-active serum. However, the difference was not statistically different from freely suspended WbtI$_{G191V}$. Three independent experiments were run in duplicate per experiment. All values represent the mean +/- the standard error of the mean. B: The presence and absence of growth after incubation of the bacteria in complement-active serum for specified amounts of time was determined for bacteria encapsulated in washed alginate or APA beads containing WbtI$_{G191V}$. Bacterial growth resulted only after one hour incubation of the bacteria encapsulated in alginate-only beads in complement-active serum. WbtI$_{G191V}$ within APA capsules grew after 48 hours of incubation in complement-active serum, similar to serum-resistant LVS, indicating an increase in protection with the additional coating around the beads. However, APA beads could not be dissolved so quantitative plate count cultures were not possible.

suspended WbtI$_{G191V}$ and LVS as positive and negative killing controls, respectively. Due to the inability to dissolve APA beads, the beads were washed, and suspended in culture medium to asses release of viable bacteria. Due to growth of the bacteria in the culture medium, accurate viable plate counts could not be completed for APA beads. Therefore, only the presence or absence of bacterial growth in broth from washed beads could be used to assess survival of WbtI$_{G191V}$ in APA beads after incubation in complement-active guinea pig serum.

Encapsulation with both the alginate beads and the APA beads increased survival of WbtI$_{G191V}$ in complement-active serum compared to freely suspended WbtI$_{G191V}$ (Fig 5). Alginate encapsulation enabled the bacteria to grow within the beads resulting in greater numbers of bacteria than at time 0, and significantly greater survival of the bacteria ($p < 0.001$) than non-encapsulated controls at 1-hour post-incubation (Fig 5A). WbtI$_{G191V}$ growth was also recorded following the 24-hour incubation of alginate beads in serum, but was not statistically greater than freely suspended WbtI$_{G191V}$ ($p = 0.128$). The addition of the PLL and alginate coating around the alginate beads enhanced survival of WbtI$_{G191V}$ to at least 48 hours post incubation, similar to LVS (Fig 5B).

## Immunization with microencapsulated WbtI$_{G191V}$ intraperitoneally stimulates greater antibody production

Groups of 4 mice each (2 male and 2 female) were immunized with WbtI$_{G191V}$ encapsulated within APA beads or freely suspended in solution to assess anti-*Francisella* antibody production. Mice were immunized either SQ or IP. Due to the size of the alginate beads IN

immunization was not possible. Only APA beads and not alginate beads were used for immunization, as $WbtI_{G191V}$ survived longer *in vitro* in complement-active serum within APA beads. Furthermore, preliminary immunization and challenge studies showed no differences in survival when mice were immunized with solely alginate beads containing $WbtI_{G191V}$ compared to mock-immunized mice (data not shown). The effect of the route of immunization on antibody production was also assessed. A low dose of LVS given SQ was used as a benchmark for antibody production. LVS was not given IP as no safe low dose exists for this route with LVS. All mice were monitored for clinical signs (*i.e.* weight loss, ruffled fur, swollen eyes, hunched back) and any moribund mice were euthanized humanely (death was not an endpoint). None of the mice immunized with $WbtI_{G191V}$ within beads or without beads showed any clinical symptoms. No swelling or vaccine-induced reactions were noted at any site of inoculation. Mice immunized with LVS SQ did exhibit ruffled fur and weight loss, but recovered in approximately 10 days.

All mice immunized with $WbtI_{G191V}$ in any of the various formulations produced an anti-*Francisella* antibody response, but only mice immunized with $WbtI_{G191V}$ with or without purified LPS in APA beads via the IP route produced antibody levels significantly higher than mock-immunized mice (Fig 6). Immunization with similar formulations of APA beads via the SQ route also produced an antibody titer, but the titers were lower than in mice immunized with LVS SQ. Due to the small sample sizes (4 mice each) and stringent statistical analysis significance is likely greater than the statistical analysis (non-parametric tests) indicated. Analysis by ANOVA with tukey's test post-hoc indicated that inoculations with all formulations (except WbtI + LPS SQ) were significantly greater than mock-immunized, with the greatest being WbtI in APA beads IP, and that all formulations inoculated IP generated a significantly greater immune response that SQ immunizations (data not shown). The raw immunization data are presented in S1 Table.

## Encapsulation of $WbtI_{G191V}$ with purified LPS protects mice against a lethal respiratory challenge

The immunized mice described above were challenged 6 weeks post-immunization with approximately $1x10^6$ CFU (5000 x $LD_{50}$) of virulent LVS IN to determine the protective efficacy of the immunization. The IN route was chosen for challenge because this route is a more realistic route of infection than IP [35]. The $LD_{50}$ of our strain of LVS is approximately 200 CFUs IN and approximately 40 CFUs IP with both routes resulting in systemic infection [36]. Mice were not immunized IP with LVS due to the very low $LD_{50}$ by this route. In comparison, the $LD_{50}$ of LVS is greater than $10^6$ CFUs SC [37]. Clinical signs were monitored and survival times post-challenge were determined (Fig 7).

Of the SQ-immunized mice, most immunized with a sublethal dose of LVS survived (75%) until the end of the study (Fig 7a). Mice immunized SQ with $WbtI_{G191V}$, encapsulated or freely suspended and with or without purified LPS, did not significantly differ in survival times to mock-immunized mice. The mean survival time was 4–5 days for all groups other than the LVS SQ-immunized group.

More mice immunized IP with $WbtI_{G191V}$ survived longer than those given the same immunization protocol SQ (Fig 7b). Mice immunized IP with $WbtI_{G191V}$ and purified LPS in APA beads had the highest survival rate at 75%, similar to mice immunized with LVS via the SQ route. The same combination of $WbtI_{G191V}$ and purified LPS freely suspended in solution resulted in 50% survival. Mice immunized with $WbtI_{G191V}$ alone encapsulated in APA beads did not survive the study period, but the mean survival time increased to 7.5 days, compared

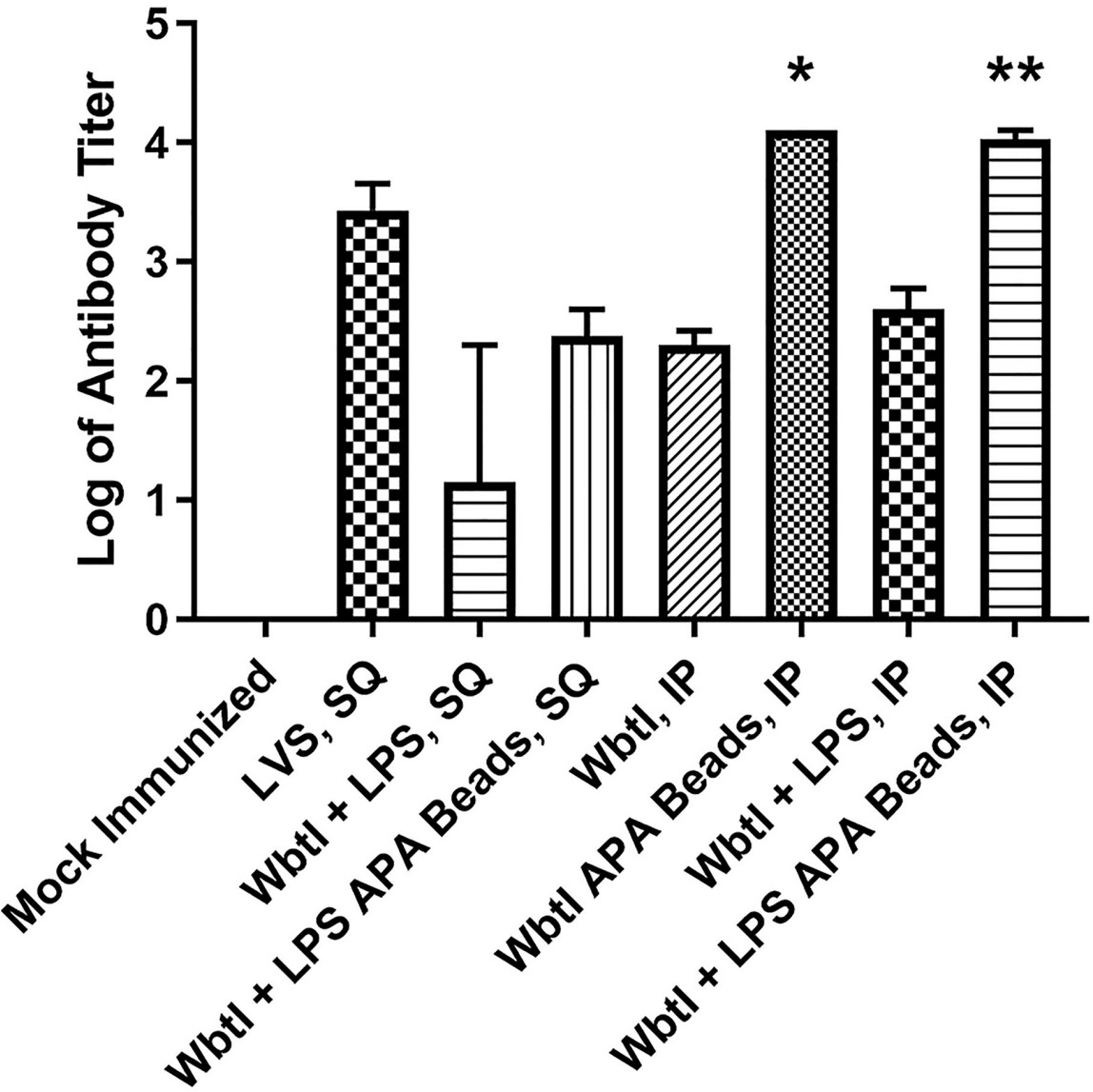

**Fig 6. Anti-LVS antibody titers in immunized mice.** Groups of 4 mice each (2 male and 2 female) were immunized with the described formulations either SQ or IP. Serum samples were taken post-immunization to determine total IgG (heavy and light chain) titers against *F. tularensis* LVS. Sera were diluted out to 1:12,800. The endpoint titer was determined to be the reciprocal of the dilution with an $OD_{450nm}$ that was 0.100 greater than the $OD_{450nm}$ of mock-immunized mice. Mock-immunized mice were inoculated IP and SQ (SQ not shown), but antibody titers were below the lowest titer measured (1:100). Bars represent the mean titer with error bars representing the standard error of the mean. Error bars are not included for mice immunized with WbtI in APA beads because all of the titers were greater than the endpoint dilution of 1:12,800. Significant antibody titer differences compared to mock-immunized mice are represented as: *: $p < 0.05$ or **: $p < 0.01$.

to 3.5 days for mock-infected mice. Mice immunized with $WbtI_{G191V}$ alone had a mean survival time of 4 days, similar to mice that were mock-immunized.

The bacterial burden of challenged mice was measured in the liver, lung, and spleen to assess protection by the various immunization protocols. No mice, regardless of the immunization protocol, fully cleared all bacteria, including mice that survived to the end of the study. Mice immunized SQ with any of the $WbtI_{G191V}$ formulations had bacterial burdens similar to mock-immunized mice mirroring the survival percentages of both groups (data not shown).

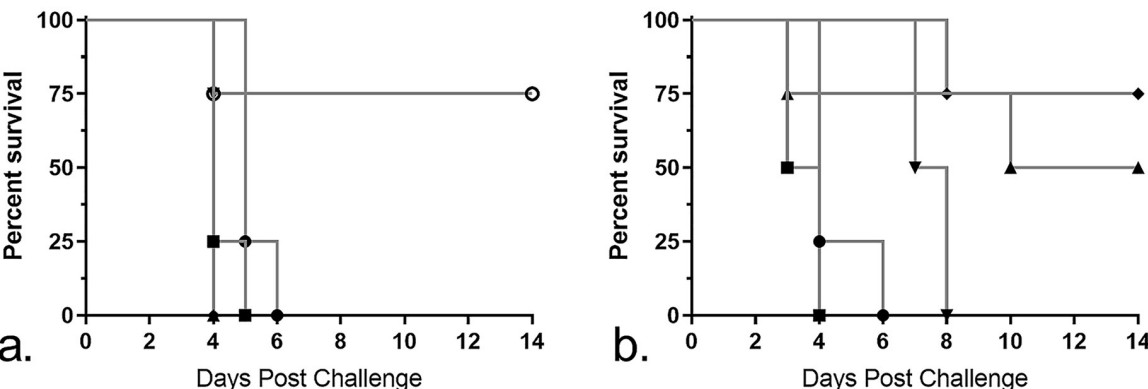

**Fig 7. Survival of mice immunized intraperitoneally and challenged with virulent LVS intranasally.** Groups of 4 mice were immunized SQ (a) or IP (b) with either WbtI$_{G191V}$ (●), WbtI$_{G191V}$ + purified LPS (▲), APA-encapsulated WbtI$_{G191V}$ (▼), APA encapsulated WbtI$_{G191V}$ + purified LPS (♦), LVS (o), or PBS (■). Six weeks post-immunization, mice were challenge with a high dose (1x10$^6$ CFU) of virulent LVS IN. Of the SQ-immunized mice, only mice that were immunized with LVS significantly survived the challenge period ($p$ = 0.0014). Of the IP-immunized mice, only mice that were immunized with WbtI$_{G191V}$ and purified LPS, either encapsulated or freely suspended in solution, survived until the end of the study. Survival was 75% ($p$ = 0.0014) and 50% ($p$ = 0.0029), respectively.

The bacterial burdens exceeded 1x10$^7$ CFU/gm of tissue in the liver, lung, and spleen for these groups. However, mice immunized SQ with LVS had significantly lower bacterial burdens than any of the other SQ-immunized groups (liver: 42 ± 31 CFU/gram of tissue, spleen: 267 ± 165 CFU/gram of tissue, and lung: 4.2 x 10$^5$ ± 3.6 x 10$^5$ CFU/gram of tissue).

Bacterial burdens were also determined for mice immunized IP (Fig 8). Mice that were immunized with WbtI$_{G191V}$ trended lower in bacterial burdens within the livers and spleens compared to mock-immunized mice (Fig 8a and 8c). Mice that survived longer tended to have lower bacterial burdens within the liver and spleen. Only mice immunized with APA beads containing WbtI$_{G191V}$ and purified LPS had statistically lower bacterial burdens than the mock-immunized mice within the liver ($p$ = 0.0245) and spleen ($p$ = 0.0348). No other immunized group was significantly lower in bacterial burden in the liver and spleen compared to the mock-immunized mice. The addition of LPS to WbtI$_{G191V}$ improved both antibody titers and lowered bacterial burden in the liver and spleen. There was no statistical difference in bacterial burden in the lungs within the IP-immunized groups.

## Mice immunized with LVS had lower histopathology scores compared to mice immunized with WbtI$_{G191V}$ in beads

After intranasal challenge with virulent *F. tularensis*, the bacteria will infect the lower respiratory tract and then spread systemically to organs such as the liver and spleen. The lung, liver, and spleen for all mice in this study were collected during necropsy and evaluated for histopathological lesions. Lungs of naïve mice exhibited moderate levels of perivascular infiltrates and mild parenchymal pneumonia. Mice immunized intraperitoneally with WbtI$_{G191V}$ and purified LPS in APA beads or freely suspended WbtI$_{G191V}$ with LPS, or LVS alone SQ had lower levels of infiltrates in the lungs, but none were statistically different from the scores for naïve mice ($p$ = 0.0629, Fig 9b). Mice immunized with LVS were the only group with a lymphocytic infiltrative population within the lungs. Within the liver and spleen, naïve mice exhibited greater histopathologic lesion scores compared to the immunized mice. In

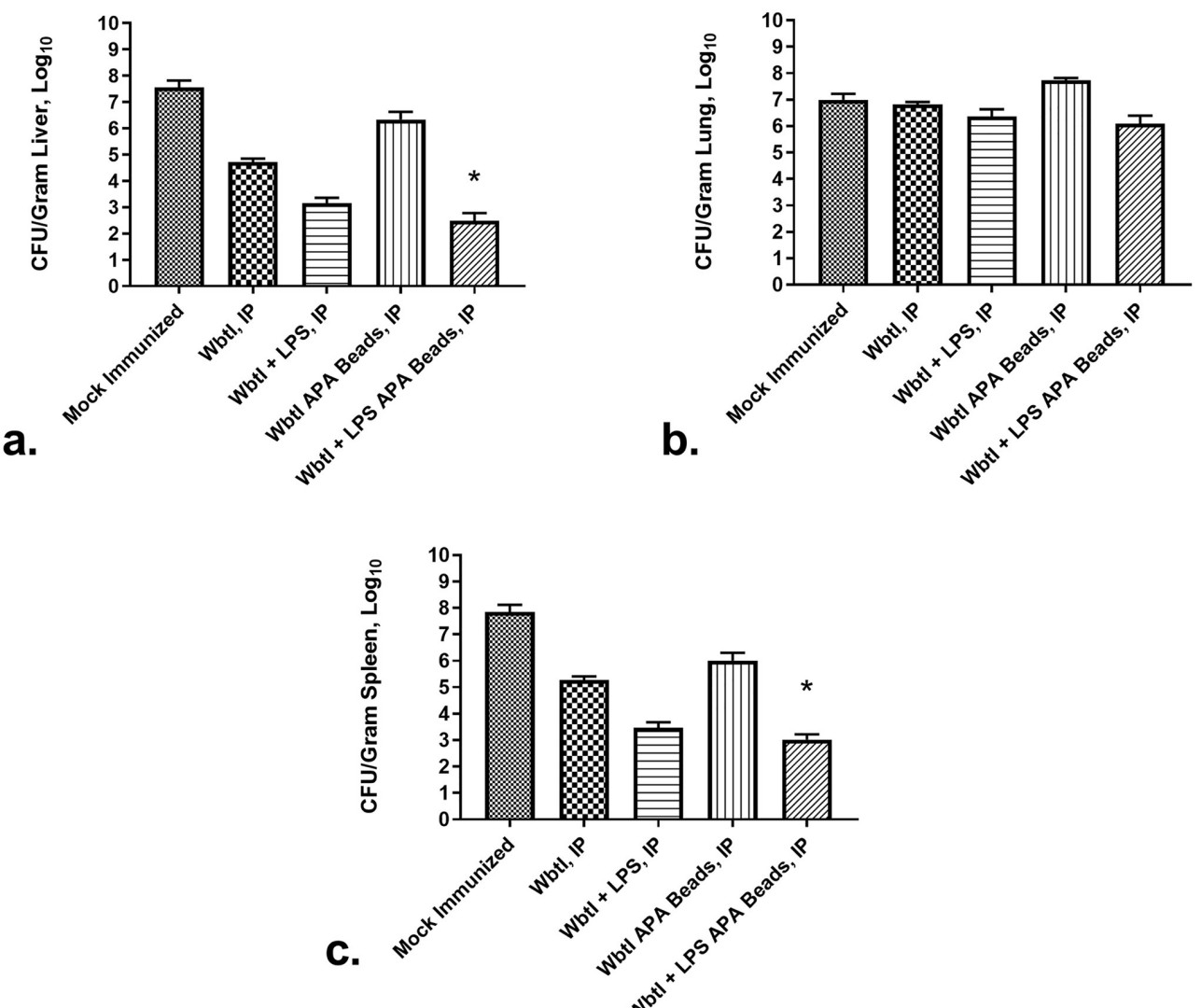

**Fig 8. Tissue bacterial loads of mice immunized intraperitoneally and then challenged with virulent LVS intranasally.** BALB/c mice (4 per IP and SQ groups) were immunized with different formulations of bacterial strains within and without APA beads, and then challenged with $1 \times 10^6$ CFU/ml of virulent LVS intranasally. At necropsy, tissue samples were harvested from the liver (a), lung (b), and spleen (c), resulting in an analysis of 42 samples read blindly. Bacterial numbers in these tissues were determined by viable plate count. Significant differences in bacterial load in a gram of tissue are indicated by: *: $p < 0.05$.

particular, immunized mice had no evidence of inflammation within the spleen compared to naïve mice ($p = 0.0110$, Fig 9c).

## Mice immunized with LVS had a greater cellular immune response following challenge than mice immunized with WbtI$_{G191V}$ in beads

Immunization of mice with LVS SQ, and WbtI$_{G191V}$ with purified LPS within APA beads IP produced similar survival rates (75% for each group) and had statistically lower bacterial burdens than the mock-immunized mice. Due to the small sample size of mice within each group (n = 4), mice immunized IP with WbtI$_{G191V}$ and purified LPS freely suspended did not have statistically different survival rates (50%) than previously mentioned groups. To further

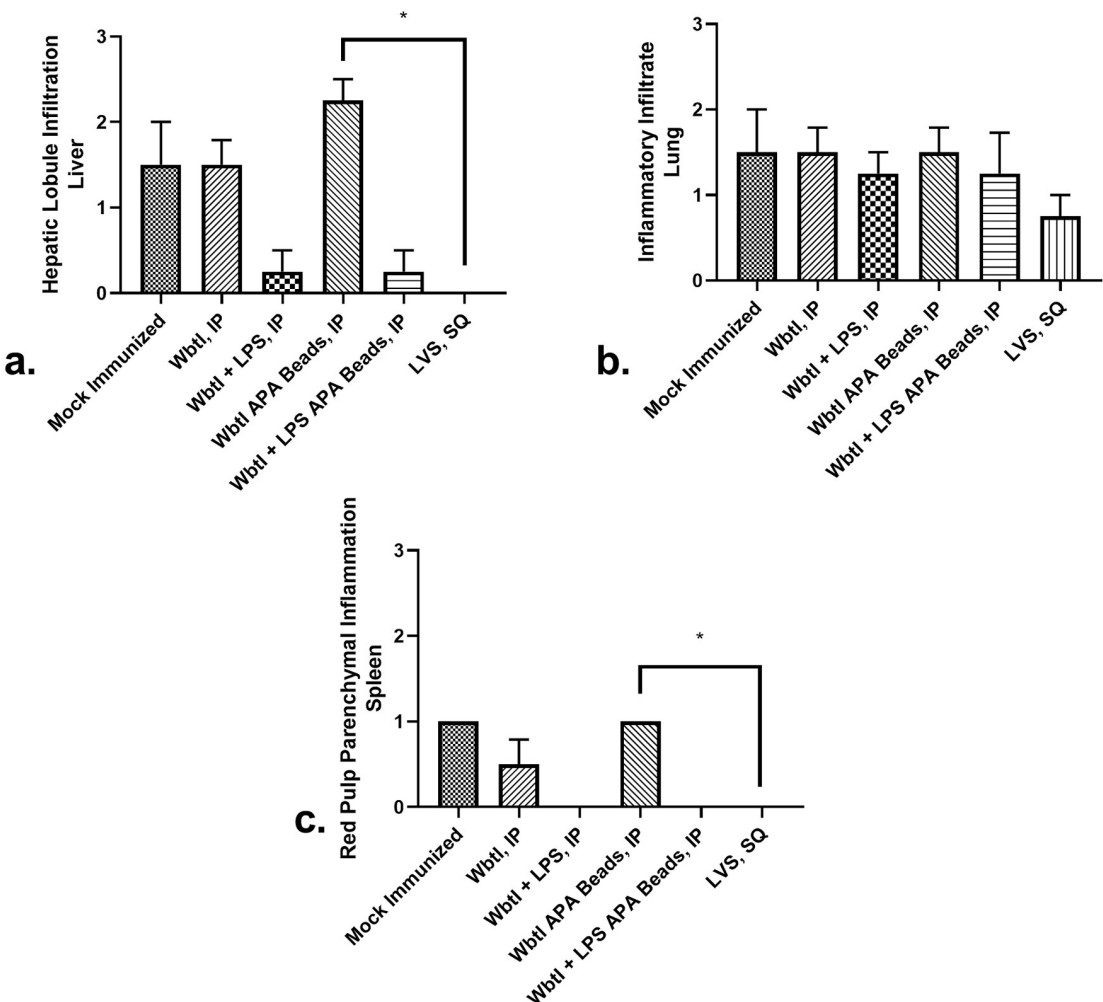

**Fig 9. Organ histopathology scores of mice immunized and then challenged with virulent LVS intranasally.** Four BALB/c mice were immunized with different formulations and then challenged with $1\times10^6$ CFU/ml of virulent LVS intranasally. All beads were APA beads. At necropsy, tissue samples were harvested from the liver (a), lung (b), and spleen (c) of all 4 mice with 3–4 samples of each tissue analyzed and read blindly, resulting in an analysis of 42 samples. Samples of tissues were fixed, stained with H&E stain, and graded for histopathological lesions. Infiltration of hepatic lobules in the liver (a) trend higher for mock immunized mice compared to immunized mice, but were not significantly different as a whole, likely due to the low number of animals used ($p = 0.0629$). Inflammatory infiltrates in the lung (b) were not significantly different between groups ($p = 0.6079$). Infiltration of red pulp in the spleen (c) were significantly lower in all immunized groups compared to the mock group ($p = 0.0110$) Significant differences in histopathology scores compared to mock immunized mice are indicated by: *: $p < 0.05$.

characterize the immune response produced within these groups, RNA was isolated from fresh frozen lung from all mice that survived challenge within these groups, and differences in gene expression related to the cellular immune response between the survivors in each group was determined. An RT$^2$ PCR Profiler Assay was used that targeted 85 genes related to the innate and adaptive immune system. A comparison of the normalized gene expression between the groups showed a greater increase in innate and adaptive immune responses of the mice immunized with LVS SQ and subsequently challenged IN with LVS compared to other immunized groups (Fig 10).

Further IPA analysis was completed to determine specific pathways affected between the immunization groups. In general, LVS-immunized mice expressed a greater level of cellular

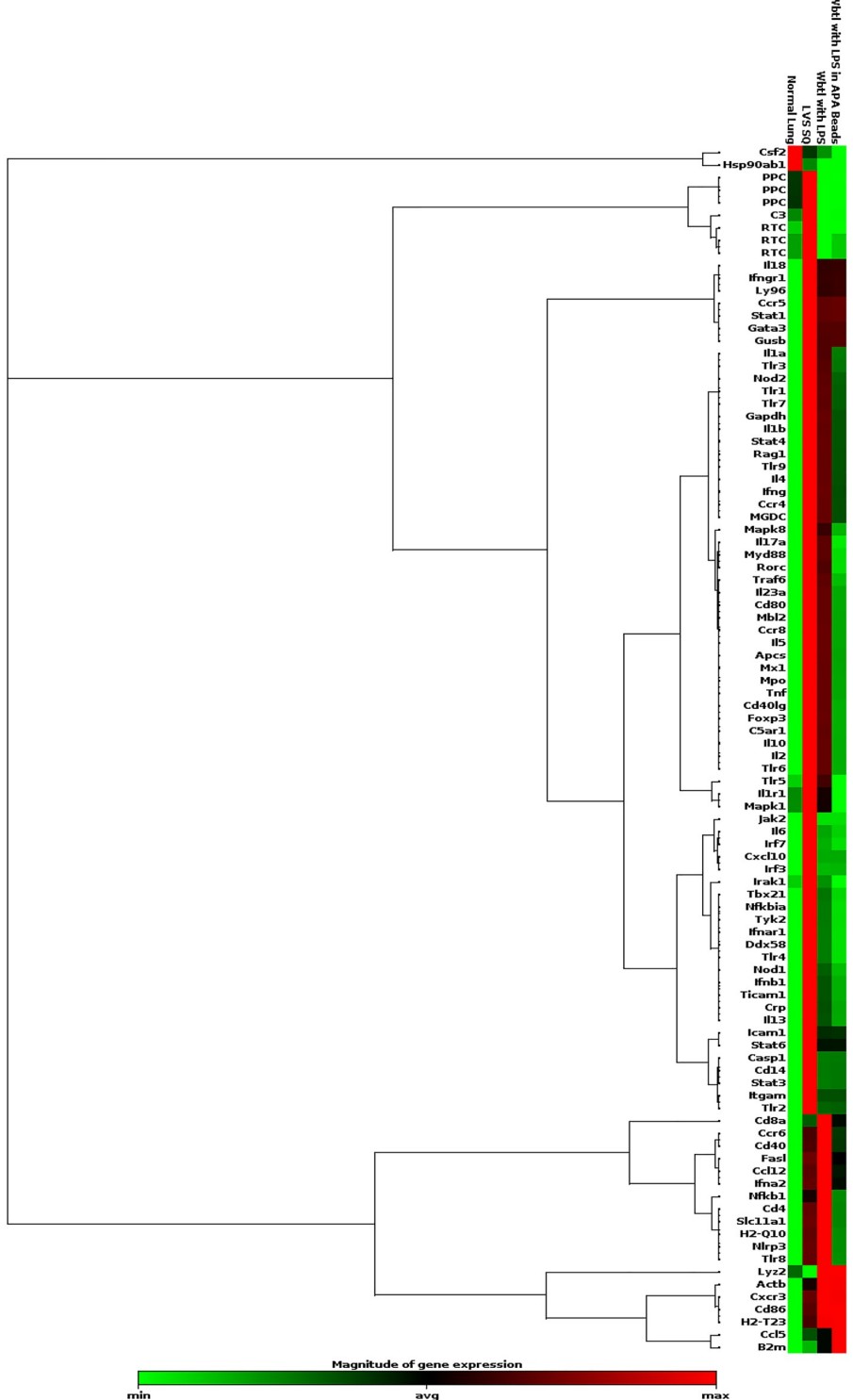

**Fig 10. Innate and adaptive immune response gene expression within lung tissue after challenge with *F. tularensis*.** Mice (four per group; 2 male and 2 female) were immunized with LVS SQ, WbtI$_{G191V}$ + LPS IP, or WbtI$_{G191V}$ + LPS within APA beads IP, and then challenged with a high dose of LVS IN. At the end of the study RNA was isolated from each lung sample of each mouse that survived challenge, then pooled by immunization group, and analyzed with an RT$^2$ PCR Profiler for the mouse innate and adaptive immune response compared to normal lung tissues. The heat map shows expression levels of each gene. Green indicates minimal expression, and red indicates maximum expression for each group and then grouped by trends in expression.

immunity following challenge than the mice immunized with WbtI$_{G191V}$ and LPS within APA beads. The cellular immune response of the group immunized with WbtI$_{G191V}$ and LPS free in suspension had a response greater mice immunized with a similar formulation in APA beads, and a lesser response than that of mice immunized with LVS only. Of particular note were the levels of interferon (IFN)-γ, TNF, and IL-6 expression. INF-γ was upregulated in all immunization groups post-challenge, but minimally upregulated in mice immunized with WbtI$_{G191V}$ and LPS in APA beads compared to the other immunization groups. IL-6 and TNF expression were similarly upregulated in all groups, but was greatest in the LVS-immunized mice. Overall, LVS-immunized mice had the highest level of INF-γ, TNF, and IL-6 expression post-challenge, which correlates with LVS-immunized mice having the lowest lung bacterial burden at the end of the study. The raw data for expression of each of the genes examined are presented in S2 Table.

## Discussion

An efficacious vaccine for tularemia is not available for public use. The previous vaccine, LVS, inadequately protects against respiratory infection with virulent Type A strains and remains partially virulent in immunocompromised individuals [38–40]. Therefore, LVS is not licensed for use as a vaccine. The most efficacious vaccine requires stimulating both cellular and antibody-mediated immunity to protect the host against the most severe route of infection: respiratory. The most effective way to stimulate both arms of the immune system is expected to be through an attenuated strain of *F. tularensis* that is safe, and adequately protective. Numerous attenuated strains of *F. tularensis* have been created [14, 38, 41, 42]. However, the efficacy of these strains against high dose respiratory challenge in humans is lacking and safety in immunocompromised patients is questionable.

Disruption of the O-antigen of the *Francisella* LPS has consistently produced strains that are highly attenuated and are highly serum sensitive [9, 10, 12, 14, 43, 44]. We theorized that an O-antigen deficient mutant of *F. tularensis* would be a very safe vaccine as no spontaneous mutation would restore the O-antigen in a mutant with one or more genes in the O-antigen locus deleted or disrupted. In this study we used an O-antigen deficient mutant of LVS with a point mutation in the *wbtI* gene, WbtI$_{G191V}$, as a model. WbtI$_{G191V}$ has been shown to be attenuated in mice to at least $2.7 \times 10^7$ CFU given IP, approximately 200,000 times the LD$_{50}$ of LVS given IP [14]. WbtI$_{G191V}$ was cleared by day 8 in mice given $1 \times 10^5$ CFU IN [14]. Mice immunized twice intradermally (ID) with the O-antigen deficient strain and challenged with $25 \times$ LD$_{50}$ of LVS IP were fully protected [14]. However, protection decreased drastically to 20% when the challenge dose of LVS was increased by a factor of 10 [14]. In order to reduce the need for a booster and increase persistence of the bacterium *in vivo*, a vaccine vehicle using alginate encapsulation was chosen.

*Brucella* species have previously been successfully encapsulated and, when used for immunization, highly protective against virulent challenges [25–27, 45, 46]. Arenas-Gamboa *et al.* demonstrated that an attenuated strain of *B. melitensis* encapsulated in APA beads was released over a one-month period *in vitro* with an initial burst release during the first few days [25]. *In vivo*, this encapsulated strain stimulated a higher level of INF-γ and antibody production compared to the non-encapsulated strain, which in theory may be due to the increased persistence of the bacterium *in vivo* within the beads [25]. As immunity to *Francisella* requires an efficient cellular immune response, this novel vaccine delivery method was attempted with WbtI$_{G191V}$ to increase persistence of the bacterium *in vivo* and, therefore, increase the cellular immune response generated. Using a similar protocol to Arenas-Gamboa *et al.* [25], WbtI$_{G191V}$ was effectively encapsulated with viable cells after completion of the procedure.

However, a thorough investigation of multiple parameters, such as nozzle size, bacterial concentration, and bead constituents has not been reported, but was examined in this study.

A sustained release vehicle could negate the need for a booster and potentially increase vaccine efficacy of a strain that is weakly protective, such as WbtI$_{G191V}$. A composite microsphere encapsulating the Hepatitis B vaccine induces an immune response comparable to the multi-dose booster series [47]. Antibody levels resulting from immunization with the alginate composite microsphere equaled or exceeded the antibody titers from the traditional two booster Hepatitis B vaccine [47]. A similar result was observed in this study with WbtI$_{G191V}$. Anti-*F. tularensis* titers for mice immunized with the alginate beads exceeded the titers of those mice immunized with freely suspended bacterial cells. Although *F. tularensis* is an intracellular pathogen, antibody-mediated protection has been shown numerous times to be important for protective immunity, possibly due to the extracellular phase of the bacterium during infection [11, 48–50]. In particular, antibodies directed to the O-antigen appear to be necessary for protection [11]. Anti-O-antigen immunity is further supported in this study based on the improved survival of mice immunized with purified LPS added to both the freely suspended WbtI$_{G191V}$ and the microencapsulated version, in comparison to the versions without purified LPS.

The alginate encapsulation procedure produces environments that are generally buffered and amendable to living cells compared to other forms of micro- or nano-encapsulation, such as liposomal procedures [18, 51]. Poly-lactic-co-glycolic acid (PGLA) has been used to create nanoparticles containing *F. tularensis* outer membrane proteins [52]. This technique can easily be used to nano-encapsulate non-living components, but creates a toxic environment during preparation for living cells. In contrast, alginate polymerization is an ionic reaction that does not require exposure of cells to extreme environments, and is desirable to encapsulate live attenuated bacteria to better stimulate a cell-mediated immune response.

*F. tularensis* was amendable to microencapsulation in sodium alginate, as shown by the visual presence of a viable GFP-labeled LVS strain. Therefore, alginate encapsulation with or without the addition of a PLL and alginate coating could effectively encapsulate live *F. tularensis* cells. Further optimization studies confirmed that the Buchi Encapsulator could be used to encapsulate *F. tularensis* cells efficiently with lower concentrations of cells ($<1\times10^8$ CFU/ml) and that an increase in the starting cell concentration decreased the encapsulation efficiency. We were unable to encapsulate cells at concentrations exceeding $1\times10^8$ CFU/ml. This phenomenon may be due to a decrease in the ionic interactions of calcium and alginate as the proportion of bacterial cells to alginate increases. This upper limit starting concentration of $1\times10^8$ cells/mL was a major limitation on the maximum inoculation dose achievable via a parenteral route of immunization.

The main goal of using an alginate vehicle was to enhance persistence of an O-antigen-deficient *F. tularensis* strain *in vivo* by providing protection against clearance mechanisms such as complement-mediated lysis. Encapsulation solely with alginate was only mildly effective at increasing survival of WbtI$_{G191V}$ in fresh serum. However, the addition of the PLL and alginate coating to create an APA bead appeared to dramatically increase serum survival. Alginate polymers are porous environments that can expand and contract with changes in tonicity. This expansion and contraction may allow complement proteins access to the encapsulated bacteria, resulting in the inefficient protection observed with WbtI$_{G191V}$ encapsulated solely in alginate. The addition of a covalently linked coating such as PLL stabilizes the beads and prevent such expansions and contraction [21]. Theoretically this would prevent access of compliment molecules to the encapsulated bacteria, but remain porous enough to allow molecules into and out of the bead in order to provide a viable environment [21]. The addition of the PLL coating increased persistence of WbtI$_{G191V}$ suspended in complement-active serum, which increased persistence of the bacteria *in vitro* and increased the antibody response to the

encapsulated cells. Therefore, WbtI$_{G191V}$ encapsulated in APA should be able to persistent *in vivo* more efficiently than non-encapsulated cells. Further studies using fluorescently labeled WbtI$_{G191V}$ to track the progress of the bacterium *in vivo* would be desirable to further study the release and time to clearance of this attenuated strain within the mouse with and without alginate encapsulation.

*F. tularensis* is an intracellular pathogen and cell-mediated immunity is highly important for protection. However, various studies have also emphasized the importance of antibody production against *Francisella* for adequate protection [3]. Cole *et al.* showed that mice immunized with purified LPS produced a significant antibody response that was protective against a virulent challenge with LVS (2–3 logs lower than the challenge dose in this study), and that the protection lasted for months after the immunization [48]. For this study, we chose to use a high level IN challenge dose ($10^6$ CFU) to stress the efficacy of the vaccine in comparison to LVS vaccination. It is likely that a lower challenge dose of $10^3$ or $10^4$ CFU would have provided protection equivalent to LVS, but would not have been an adequate comparison at the protection limit. Furthermore, antibody production alone did not correlate strongly with successful high-challenge protection. WbtI$_{G191V}$ combined with purified LPS encapsulated in APA beads produced a significant antibody response following IP immunization and 75% of these mice survived a virulent high-dose IN challenge. However, a similar formulation without the purified LPS also produced a significant antibody response following IP immunization that did not confer protection against a virulent challenge. In comparison, LVS immunization SQ produced a smaller, non-statistically significant antibody response (significance may have been affected by the low numbers of animals used) in mice and 75% of these mice survived a virulent IN challenge. Though mice with higher anti-*Francisella* antibody titers tended to survive longer post-challenge in this study, an exact correlation between antibody titer and protection could not be made. The presence of antibodies to *Francisella* may still be a contributing factor for protection. Stenmark *et al.* showed that mice immunized with LVS given immune serum prior to challenge were not protected any better than mice solely immunized with LVS; but that naïve mice and B-cell deficient mice administered immune serum prior to challenge did survive lethal doses of LVS [53]. In addition, vaccination with outer membrane proteins suspended in PGLA nanospheres induced significant antibody titers and protected mice against a virulent LVS challenge, but did not protect mice against challenge with SchuS4 [52]. However, when mice immunized with the PGLA nanospheres were then boosted with live LVS the protection against the SchuS4 challenge increased [52]. A true infection, which is somewhat mimicked by repeated immunization with LVS, may be necessary for a tularemia vaccine to induce the necessary cell-mediated immune response, and not just an antibody-mediated response. LVS is considered virulent in mice and will produce a more robust infection than WbtI$_{G191V}$. Even encapsulated, WbtI$_{G191V}$ may be too attenuated to persist long enough to induce the necessary cell-mediated immune response, but did help stimulate antibody-mediated immunity resulting in the partial protection exhibited here.

Alginate encapsulation may also be impeding the development of cell-mediated immunity, as the lowest cell-mediated immune response was found in mice immunized with alginate-encapsulated cells. The *Brucella abortus* vaccine, RB51, when encapsulated in alginate stimulates an effective Th1 immune response in deer that was statistically elevated compared to RB51 alone [27]. Ajdary *et al.* also found that alginate encapsulation of the *Mycobacterium tuberculosis* vaccine Bacille Calmette-Guerin (BCG) increased the cell-mediated immune response and specifically the production of IFN-γ [54]. However, in this study immunization of mice with WbtI$_{G191V}$ encapsulated in alginate in the APA form produced the lowest expression of IFN-γ associated pathways, which is essential for protection against wild-type *F. tularensis* challenge [55, 56].

With an intranasal route of inoculation, infection with LVS spreads from the lungs to other portions of the reticuloendothelial systems, such as the liver and spleen. Mice infected with virulent LVS demonstrate interstitial edema of the lungs, mild lymphocytic depletion and macrophage infiltration of the spleen, and areas of inflammatory infiltrates within the liver [57, 58]. Mock-immunized mice were affected in a similar fashion in this study, and compared to immunized mice, showed more severe lesions by histopathology. LVS-immunized mice had a prominent lymphocytic infiltrate in the lungs compared to any of the other groups. T lymphocytes have been repeatedly shown to be important in controlling *F. tularensis* infections [55, 56]. B lymphocytes have also been shown to internalize *F. tularensis* cells during infection and produce a variety of cytokines, including IFN-γ and tumor necrosis factor (TNF) [59, 60]. IFN-γ and TNF were upregulated strongly in LVS-immunized mice compared to the other immunization groups in this study.

In conclusion, this study identifies parameters to be examined for optimal alginate encapsulation of bacteria and demonstrates that alginate encapsulation of an attenuated *F. tularensis* strain can protect the bacterium from the host immune system and stimulate an improved antibody-mediated immune response. However, the alginate encapsulation appeared to dampen the ability of the live attenuated strain to induce the essential cell-mediated immune response. The addition of a protein conjugate that stimulates a Th1 response to the outer shell of the alginate microsphere, or the use of an attenuated strain that is not cleared as quickly, could increase the ability of this immunization to induce a more robust cell-mediated immune response.

## Supporting information

**S1 Table.**
(XLSX)

**S2 Table.**
(XLSX)

## Acknowledgments

We would like to thank Armaghan Nasim for valuable assistance with viable plate counts and the animal care staff at the Center for One Health Research at the VMCVM for their assistance in caring for and handling the animals used in this project. We thank Martha Furie for providing LVS expressing Green Fluorescent Protein.

## Author Contributions

**Conceptualization:** Thomas J. Inzana.

**Data curation:** Kelly C. Freudenberger Catanzaro, Kevin K. Lahmers, Irving C. Allen.

**Formal analysis:** Kelly C. Freudenberger Catanzaro, Kevin K. Lahmers, Irving C. Allen, Thomas J. Inzana.

**Funding acquisition:** Thomas J. Inzana.

**Investigation:** Kelly C. Freudenberger Catanzaro, Irving C. Allen, Thomas J. Inzana.

**Methodology:** Kelly C. Freudenberger Catanzaro, Kevin K. Lahmers, Irving C. Allen, Thomas J. Inzana.

**Project administration:** Thomas J. Inzana.

**Resources:** Thomas J. Inzana.

**Software:** Irving C. Allen.

**Supervision:** Thomas J. Inzana.

**Validation:** Kelly C. Freudenberger Catanzaro, Irving C. Allen, Thomas J. Inzana.

**Writing – original draft:** Kelly C. Freudenberger Catanzaro.

**Writing – review & editing:** Kelly C. Freudenberger Catanzaro, Kevin K. Lahmers, Irving C. Allen, Thomas J. Inzana.

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
