## [Decision Letter · Decision Letter 0]

14 Dec 2021

PONE-D-21-33517Alginate microencapsulation of an attenuated O-antigen mutant of Francisella tularensis LVS as a model for a vaccine delivery vehiclePLOS ONE

Dear Dr. Inzana,

Thank you for submitting your manuscript to PLOS ONE. After careful consideration, we feel that it has merit but does not fully meet PLOS ONE’s publication criteria as it currently stands. Therefore, we invite you to submit a revised version of the manuscript that addresses the points raised during the review process.

Both reviewers emphasized that the approach detailed in the manuscript may represent a novel vaccine delivery platform and may be adapted to any other vaccine formulation. Reviewer 1 has expressed some minor concerns. However, reviewer 2 raised serious concerns about some of the overstated conclusions not supported by the data, the reproducibility of the data as it is not clear that how many times the in vivo experiments were repeated, and a low number of animals used in the experiments. These concerns must be addressed to interpret the results accurately, and the conclusions are required to be modified accordingly.

Please submit your revised manuscript  and If you will need more time than this to complete your revisions, please reply to this message or contact the journal office at plosone@plos.org. Please include the following items when submitting your revised manuscript:A rebuttal letter that responds to each point raised by the academic editor and reviewer(s). You should upload this letter as a separate file labeled 'Response to Reviewers'.A marked-up copy of your manuscript that highlights changes made to the original version. You should upload this as a separate file labeled 'Revised Manuscript with Track Changes'.An unmarked version of your revised paper without tracked changes. You should upload this as a separate file labeled 'Manuscript'.

We look forward to receiving your revised manuscript.

Kind regards,

Chandra Shekhar Bakshi, DVM, Ph.D.

Academic Editor

PLOS ONE

Journal Requirements:

“This work was supported by the Tyler J. and Frances F. Young Foundation, the Virginia-Maryland College of Veterinary Medicine, and the Stamps Family Charitable Foundation.”

“K F-C was supported by the Stamps Family Charitable Foundation (no fund number).

TJI was supported by the Tyler J. and Frances F. Young Foundation, and the Virginia-Maryland College of Veterinary Medicine (no fund number).”

4. Please ensure that you refer to Figure 1 in your text as, if accepted, production will need this reference to link the reader to the figure.

Reviewers' comments:

Reviewer's Responses to Questions

**Comments to the Author**

1. Is the manuscript technically sound, and do the data support the conclusions?

Reviewer #1: Yes

Reviewer #2: Partly

2. Has the statistical analysis been performed appropriately and rigorously? 

Reviewer #1: Yes

Reviewer #2: No

3. Have the authors made all data underlying the findings in their manuscript fully available?

Reviewer #1: Yes

Reviewer #2: No

4. Is the manuscript presented in an intelligible fashion and written in standard English?

Reviewer #1: Yes

Reviewer #2: Yes

5. Review Comments to the Author

Reviewer #1: This is well-written manuscript detailing an interesting method (APA encapsulation) of vaccine delivery that enables a "Ag-depot effect".

A few minor issues/typos to be fixed are listed below:

Pg 4. Mentions of the genetic instability of LVS are unnecessary as the manuscript does not mention or show that the wbtI mutant of LVS is more or less stable.

pg 15 bold font

In the figure legends, I would remove the P value definitions that do not apply to a given figure.

Reviewer #2: PONE-D-21-33517

In this paper, by Freudenberger Catanzaro et al, the authors describe the alginate encapsulation of an LVS-attenuated strain as a vehicle to extend the presence of the vaccine in vivo and, therefore, to induce a stronger immune response than the non-encapsulated formulation. The optimization of the method is well described and allowed the authors to successfully obtain an alginate and an APA formulations of F. tularensis. However, in contrast to the literature reported by the authors, this approach seems to inhibit T cell immune responses, which is essential against F. tularensis.

These information, positive and negative, are worthy of publication. However, some conclusions, related to the in vivo studies, may be overstated and not completely supported by the data. The major problem is that it is not clear how many times the experiments were repeated and therefore whether they are reproducible. For this reason, conclusions related to the Abs production, survival, organ burden and pathology may be overstated if the experiment was performed only once and if the number of animals used in the study was small. For example, Fig 6 shows higher titers for the APA vaccines, but apparently that is not statistically different from the non-encapsulated vaccines, likely because the number of mice was small or, because, in contrast to the authors’ statement, there is really no difference. My recommendation is to tune down the conclusions derived from experiments performed only once and present the data as preliminary, showing a trend that needs further investigation.

Major

Fig 5A. Overall the figure does not add more information than the table (5B), but adds some confusion. For example, at 1 hour the encapsulated survival is above 100%; at 1 hour, it is not clear how the standard deviation was calculated and why there is not standard deviation at 0 hrs. The author may consider removing the figure. Otherwise, in the text please explain how the survival can be above 100%. In the figure, in addition to the asterisks, add a bracket indicating which columns were statistically different. In the legend, specify whether “encapsulated” refers to alginate or APA; indicate how many samples were used to calculate the survival. Please also indicate what the values represent (mean, median, standard deviation, SEM or else) and why there is no standard deviation at 0 hrs.

Fig 6 and legend. It appears that all vaccine groups produce Abs. However, most, including LVS, are not statistically different than the mock immunized, which is unexpected. To better understand the extend of this quantification and of the statistical analysis, please indicate how many mice were tested for each vaccine group, including the mock group. In addition, Wbtl APA beads IP does not have a standard curve: is it because only 1 mouse was used? Is the value of the mock immunized really 0? Was the immunization of the mock SQ, IP or both? If both, the data of one of the two is missing. Similarly, the data for the mice immunized with Wbtl and Wbtl APA SQ are missing. To fully evaluate whether the APA vaccines produce more Abs than the non-APA, please include information on these two groups. In addition, please indicate whether the data is a representative of multiple experiments, and eventually how many times was the experiment repeated. Moreover, indicate what the values represent (mean, median, etc). The authors state that the APA vaccines produce more Abs than the non-encapsulated: within each vaccination route, SQ and IP, are the titers from the APA vaccine groups statistically different from the non-APA vaccines?

Fig 7 and legend. It appears that the same mice described in Fig 6 were challenged and their survival described in this figure. Therefore, similar to the previous comments and to better understand the significance of these data, please indicate how many mice were used for each vaccine group, how many times the experiment was repeated and whether the data are from a representative experiment or else.

Fig 8. As above, indicate number of mice, how many times the experiment was repeated, whether the data are from a representative experiment, and what the values represent (mean or median, St deviation, SEM).

Page 25, 13-21, Fig 9 and legend; the results described on page 25 (mice immunized SQ, lung in panel 9a, etc.) do not correspond with the data presented in the figure (IP vaccination, liver in panel 9a, etc.). In addition, indicate number of mice that were analyzed, what the values represent, how many times the experiment was repeated. Moreover, the data from Wbtl freely suspended and Wbtl APA Beads are missing. Please describe their results and add them in the figure.

Fig 10 legend. Indicate how many mice were used in the experiment, whether the RNA was pooled from different animals or whether individual RNA from multiple animals was analyzed by PCR and their results combined.

Minor

Page 4 lines 2-3. “potential” and “possible” are redundant. Consider editing as: “potential use of this bacterium as bioweapon”.

Pag 15, line 4: Please include the reference to Fig 1 (“reference not found”).

Pag 17, lines 3-4 and Fig 2. Test indicates that the differences were designates with asterisks. However, Fig 2 does not show any asterisks or brackets. Please edit the figure to include the asterisks.

Page 17, lines 6 and 16 and page 18, line 17. Please edit the wide space preceding ”Fig 3”.

Fig 3: Consider simplifying the title of the x and y axes such as “starting CFU/ml” and CFU/ml of beads (log10)”.

Page 11, line 16. It is stated that mice were immunized IP, SQ or orally. However, in page 21, line 11 it is stated that mice were immunized only SQ or IP, not orally. Please explain and/or edit accordingly.

Page 24, lines 6-14; please indicate that the data for the SQ are not shown or, alternatively, show the data.

Fig 6, legend and corresponding materials and method. Please indicate which antibodies have been detected: total IgG, IgM, IgG subtypes.

Fig 8. Although it has been described in the legend, for easy reading, include spleen, liver and lung as title of the corresponding panel or as y-axes title (e.g. CFU/gr Lung (log10)). Moreover, to avoid confusion, the name of the vaccines should be consistent with the previous figures and with the test.

Fig 9. For easy reading, include spleen, liver and lung as title of the corresponding panel. Moreover, the name of the vaccines should be consistent with the previous figures and with the test.

Page 28, line 13; the authors indicate that “LVS is no longer licensed”. Do the authors refer in the US, which apparently has never been licensed, or other countries? Please clarify and edit accordingly.

6. PLOS authors have the option to publish the peer review history of their article (what does this mean?). If published, this will include your full peer review and any attached files.

Reviewer #1: No

Reviewer #2: No

---

## [Author Response · Author response to Decision Letter 0]

11 Feb 2022

Response to reviewers

Reviewer 1

Pg. 4: revised as suggested

Pg 15: Corrected.

P values that are not applicable have been removed from the figure legends.

Reviewer 2

The number of times key experiments were done have been added as well as clarification of the number of animals used.

Major points

Fig. 5A. The reason survival is >100% at 1 hour incubation is because the bacteria grew in highly bactericidal serum, indicating they were well protected by the beads. Each bar represents the mean + the standard error of the mean (SEM). Clarification of how SEMs were calculated and the number of experiments done has been added. Three independent experiments/replicates were run in duplicate per time point (3 separate times with duplicates (mechanical replicates) for each). There are no SEMs at 0 hours because 100% of the bacteria were viable in all experiments and groups t that time. Asterisks and brackets have been added where relevant. Three independent experiments were run in duplicate per experiment. This has been added to the Materials and Methods and to the legend. The figure is more quantitative than the table. However, the APA beads could not readily be dissolved like the alginate-only beads, and therefore is qualitative only. Therefore, we could only incubate the beands in complement-active serum for the indicated period and assay for viable bacteria. Fig. 5A shows the quantitative results for alginate beads and 5B also shows the qualitative results for bacteria within APA beads. Fig. 5B shows that the bacteria could survive longer in the APA beads than in alginate alone (24 and 48 hrs). Therefore, we would prefer to keep the revised Fig. 5A and 5B as is. Encapsulated in Fig. 5A refers to alginate alone, and has been clarified in the legend. 

Fig. 6 and legend. I presume the reviewer is referring to LVS here as the “mock immunized”, but the “mock immunized” mice received only buffer and made no antibody response. The number of mice used in these vaccine trials have now been added to the Methods section and the legend. Four mice (2 male and 2 female) were used in each vaccine group is order to keep the numbers of mice used to the minimum number that can yield statistically significant results, as required by IACUC. The number of mice that were not immunized (mock) were 2, also to keep the number of control mice used to a minimum. Four mice were immunized with WbtI APA beads, IP. There is no SD on this column because all the mice made antibody titers greater than 1:12,800 and that was as far as the sera was tittered out. This information has been added to the legend. The mock-immunized mice were inoculated IP. Some mock-immunized mice were inoculated SQ, but these data were not shown because there was no difference from those inoculated IP. I cannot say that the titers of the mock-immunized mice were 0, but they were below the level of lowest dilution (1:100). This information has been added to the legend. The antibody titers of mice inoculated with WbtI and WbtI APA SQ were not determined. LVS was not inoculated IP because there is no sublethal dose for that strain via that route. However, the emphasis of this figure is on the comparison of how the addition of LPS or LVS in APA beads affects the immune response. The data are representative of 3 separate experiments carried out in duplicate, so each value is the mean of 6 experiments. The values represent the mean titer + the standard error of the mean. The APA vaccine groups are not statistically different from the non-APA vaccine group (at least WbtI+LPS vs WbtI+LPS APA are not significantly different titer-wise). The only significant difference is between the Mock vs WbtI+LPS and Mock vs WbtI+LPS APA when the analysis was done using the more rigorous non-parametric analyses due to the relatively small number of individuals/ group (4). However, when reanalyzed using ANOVA with tukey’s test post-hoc all formulations were significantly greater than mock-immunized, except WbtI + LPS SQ, and all formulations inoculated IP were significantly greater than the same formulations inoculated SQ. As mentioned in the text, the immune response to F. tularensis within alginate only beads was not measured because WbtIG191V survived significantly longer in beads with the APA coating. As we were seeking the optimal immune response and use of the minimal number of animals, non-APA vaccines were not examined. 

Fig. 7. As stated above, each vaccine group consisted of 4 mice (2 male and 2 female). The experiment was not repeated because the results were not that dramatically different. The data are representative of the experiment done.

Fig. 8. The histopathology consisted of 4 mice per IP and SQ groups, resulting in 42 samples analyzed, and read blindly. This information has been added to the Methods section and the legend.

Page 25, 13-2; Fig. 9 and legend: There was a typo in this sentence and “subcutaneously” has been changed to intraperitoneally” in the results, and inoculation with LVS clarified to be SQ. The reference to “9a” has been corrected to “9b”. “9b” has been corrected to “9c”. Fig. 9 has been redone and additional data have been added to it. Three-to four samples were analyzed and read of each tissue from each of the 4 mice, resulting in 42 total samples being read and analyzed. This information has been added to the legend. The values are represented on a scale of 0-3, 0 = no inflammatory infiltrates present. Grading of 1-3 represent increasing amounts of cellular infiltrates and are based on two references that have been added to the manuscript. This information has been added to the Methods section.

Fig. 10 legend: As described in the Methods section, RNA was isolated from each sample of each animal, and then pooled by immunization group. This information has been added to the legend.

Minor points

Pg. 4: Revised as suggested.

Page 15, line 4: There is no reference needed at this specific site. The reference to Fig. 1 has been moved to where it is relevant.

Page 17, lines 3-4; Fig 2.: Fig. 2 has been revised to include asterisks and brackets to show difference in significance.

Page 17, line 6, 16 and page 18, line 17: spacing around “Fig. 3” has been corrected.

Fig. 3: titles of x and y axis have been revised as suggested.

Page 11, line 16: oral immunization was planned originally , but due to technical problems was not done. Mention of oral immunization has been taken out of the methods section.

Page 24, lines 6-14: Although the data are reported in the text, the statement “data not shown” has been added, as the data are not shown in a table or figure.

Fig. 6 legend and M&M: We have added that the antibodies detected are total IgG (H&L chains).

Figs. 8 and 9: “spleen, liver and lung” have been added to the appropriate panels and the terms for the material used for immunization has been made consistent with that in Fig. 6.

Page 28, line 13: That LVS is no longer licensed has been changed to LVS is not licensed.

---

## [Decision Letter · Decision Letter 1]

24 Feb 2022

Alginate microencapsulation of an attenuated O-antigen mutant of Francisella tularensis LVS as a model for a vaccine delivery vehicle

PONE-D-21-33517R1

Dear Dr. Inzana,

We’re pleased to inform you that your manuscript has been judged scientifically suitable for publication and will be formally accepted for publication once it meets all outstanding technical requirements.

Kind regards,

Chandra Shekhar Bakshi, DVM, Ph.D.

Academic Editor

PLOS ONE

Additional Editor Comments (optional):

Reviewers' comments:

Reviewer's Responses to Questions

**Comments to the Author**

1. If the authors have adequately addressed your comments raised in a previous round of review and you feel that this manuscript is now acceptable for publication, you may indicate that here to bypass the “Comments to the Author” section, enter your conflict of interest statement in the “Confidential to Editor” section, and submit your "Accept" recommendation.

Reviewer #2: All comments have been addressed

2. Is the manuscript technically sound, and do the data support the conclusions?

Reviewer #2: Yes

3. Has the statistical analysis been performed appropriately and rigorously? 

Reviewer #2: Yes

4. Have the authors made all data underlying the findings in their manuscript fully available?

Reviewer #2: Yes

5. Is the manuscript presented in an intelligible fashion and written in standard English?

Reviewer #2: Yes

6. Review Comments to the Author

Reviewer #2: (No Response)

7. PLOS authors have the option to publish the peer review history of their article (what does this mean?). If published, this will include your full peer review and any attached files.

Reviewer #2: No

---

## [Editor Report · Acceptance letter]

2 Mar 2022

PONE-D-21-33517R1 

Alginate microencapsulation of an attenuated O-antigen mutant of *Francisella tularensis* LVS as a model for a vaccine delivery vehicle 

Dear Dr. Inzana:

I'm pleased to inform you that your manuscript has been deemed suitable for publication in PLOS ONE. Congratulations! Your manuscript is now with our production department. 

Kind regards, 

on behalf of

Dr Chandra Shekhar Bakshi 

Academic Editor

PLOS ONE